# Better late than never: Optimising the proteomic analysis of field-collected octopus

**Qiaz Q. H. Hua**[1]*, **Clifford Young**[2,3], **Tara L. Pukala**[4], **Jasmin C. Martino**[2], **Peter Hoffmann**[2,3], **Bronwyn M. Gillanders**[1], **Zoe A. Doubleday**[2]*

**1** Environment Institute and School of Biological Sciences, University of Adelaide, Adelaide, South Australia, Australia, **2** Future Industries Institute, University of South Australia, Mawson Lakes, South Australia, Australia, **3** Clinical & Health Sciences, University of South Australia, Adelaide, South Australia, Australia, **4** Department of Chemistry, School of Physics, Chemistry and Earth Sciences, University of Adelaide, Adelaide, South Australia, Australia

* qiaz.qh.hua@gmail.com (QQHH); zoe.doubleday@unisa.edu.au (ZAD)

**Data Availability Statement:** Data are available via ProteomeXchange with identifier PXD043096.

**Funding:** This work was supported by the Future Industries Accelerator Scheme, University of South

## Abstract

Proteomics, the temporal study of proteins expressed by an organism, is a powerful technique that can reveal how organisms respond to biological perturbations, such as disease and environmental stress. Yet, the use of proteomics for addressing ecological questions has been limited, partly due to inadequate protocols for the sampling and preparation of animal tissues from the field. Although RNA*later* is an ideal alternative to freezing for tissue preservation in transcriptomics studies, its suitability for the field could be more broadly examined. Moreover, existing protocols require samples to be preserved immediately to maintain protein integrity, yet the effects of delays in preservation on proteomic analyses have not been thoroughly tested. Hence, we optimised a proteomic workflow for wild-caught samples. First, we conducted a preliminary in-lab test using SDS-PAGE analysis on aquaria-reared *Octopus berrima* confirming that RNA*later* can effectively preserve proteins up to 6 h after incubation, supporting its use in the field. Subsequently, we collected arm tips from wild-caught *Octopus berrima* and preserved them in homemade RNA*later* immediately, 3 h, and 6 h after euthanasia. Processed tissue samples were analysed by liquid chromatography tandem mass spectrometry to ascertain protein differences between time delay in tissue preservation, as well as the influence of sex, tissue type, and tissue homogenisation methods. Over 3500 proteins were identified from all tissues, with bioinformatic analysis revealing protein abundances were largely consistent regardless of sample treatment. However, nearly 10% additional proteins were detected from tissues homogenised with metal beads compared to liquid nitrogen methods, indicating the beads were more efficient at extracting proteins. Our optimised workflow demonstrates that sampling non-model organisms from remote field sites is achievable and can facilitate extensive proteomic coverage without compromising protein integrity.

## Introduction

Understanding a species response to external biological perturbations such as diseases and environmental stress can be achieved by studying the underlying molecular mechanisms.

Australia to ZAD (grant no. FIA049). QQHH is
supported by the University of Adelaide Research
Scholarship. The funders had no role in study
design, data collection and analysis, decision to
publish, or preparation of the manuscript.

**Competing interests:** The authors have declared
that no competing interests exist.

Proteomics is a discipline that examines the functional products of temporal gene expression (proteins), but has not been extensively applied in ecology [1]. Proteomics allows the examination of changes in protein quantities and their post-translational modification status [2]. While amino acid sequences may reveal biological function and origin [3], functional analysis of the proteome yields information on important physiological mechanisms, such as cellular stress responses [4]. Proteins that are expressed at any given time reveal information on the cellular state and active cellular pathways [3]. Technological advances in mass spectrometry technology and data search programs coupled with the increasingly affordable sequencing technology are also opening new research avenues for proteomics data, such as examining environmental stress, adaptation strategies, immune defence mechanisms, seafood provenance, venomics, and complementing genomic findings [2, 3, 5–8].

Despite its clear advantages and broad applications, there is a relative lack of ecological studies that use proteomics to understand human impacts and stressors, such as global warming and ocean acidification. This is partly attributed to the shortage of protocols optimised for field-collected organisms. The first step to any successful proteomic analyses is obtaining and preserving tissues without compromising protein integrity. Thus, existing protocols require samples to be preserved immediately. However, this is often impractical for field-based ecological applications. For instance, field sampling sites may be remote and sampling may take place within awkward, confined spaces, such as a moving boat, where only ice may be available. This greatly limits the ability to follow standard protocols of immediate preservation, and challenging fieldwork conditions often mean delays in preserving tissues. However, the effects of such time delays in tissue preservation on proteomic analysis has not been extensively examined. In addition to this time factor, the standard practice of preserving tissues in liquid nitrogen (LN$_2$) or freezing temperatures (-20˚C or -80˚C) is not practical given that most field locations lack the required infrastructure. The use of LN$_2$ and ethanol also poses a safety concern in the field especially for long-distance travel or in the confined spaces of a vehicle. These necessitate safer and more practical alternatives such as the use of the storage reagent RNA*later*, which can successfully preserve DNA, RNA and proteins [9, 10].

To date, proteomic approaches have been used to study environmental impacts on mussels [11], gobies [12], crabs [13], sea squirts [14], oysters [15], plants [16], limpets [17] and zebrafish [18], and venom characterisation such as in snakes [19] and octopus [20]. However, these studies have largely been performed using captive organisms, highlighting the fact that ecological proteomics remains limited due to a lack of established protocols for field-collected samples, especially for tough muscular tissues present in octopuses and other molluscs. Proteomics has enormous potential to be applied in more marine and terrestrial organisms, especially in the face of climate change, but establishing protocols to allow proteomic analysis of field-collected organisms needs to be first developed and tested.

In this study, we optimised proteomic methods for both sample preservation and preparation using remote field-caught octopus. To fulfil our aims of method optimisation, we examined the arm proteome as octopus arms can be rapidly sampled both in the laboratory and field compared to other tissues such as organs. Given that existing protocols typically use soft tissues, muscular tissues like octopus arms require further optimisation to ensure efficient protein extraction. We first determined the suitability of RNA*later* to maintain protein integrity using fresh octopus tissues preserved according to standard protocols. Next, we conducted comparative proteomic analyses on wild-caught octopus tissues obtained from field settings and compared among different tissue types, sexes, tissue homogenisation methods and time delay in preservation. By comparing the number of proteins identified and their relative abundances between samples, we obtained key information that provides recommendations for the future proteomic analyses of wild specimens.

## Materials and methods

### Optimisation of test samples with RNA*later*

Three female *Octopus berrima* (biological replicates) were collected directly from a commercial fishery in South Australia and raised in aquaria for three months until spawning and hatching of embryos. They were then euthanised in the lab in accordance with the Australian code for the care and use of animals for scientific purposes [21] by exposure to 1.5% magnesium chloride for 10 min, followed by exposure to 3.5% magnesium chloride for 30 min. Arm sections were dissected and immediately stored at -20°C, which is a recommended temperature for storage of proteomics samples and these samples were hence used as the control in this study. To determine if RNA*later* incubation helped maintain protein integrity, we compared samples from three timepoints: immediately, after 3 h and 6 h in the presence or absence of RNA*later*, during which samples were maintained at 4°C. These timepoints were based on the time necessary to complete the fieldwork. We then compared the effect of homemade (S1 Table) and commercial RNA*later* (Thermo Fisher Scientific, Waltham, USA) on protein quality and quantity. We additionally compared these samples maintained at 4°C without RNA*later* with those incubated at room temperature without RNA*later* to ascertain if cold storage was sufficient for maintaining protein quality. For samples with RNA*later*, 5X volume of RNA*later* was added to each arm section for the durations stated above.

Thawed tissues (approximately 10 mg each) were prepared according to the above conditions before being transferred into a 2 ml tube containing 200 μl RIPA buffer and four 50 mg metal beads for tissue lysis on the Precellys Evolution homogeniser (Bertin Technologies, Montigny-le-Bretonneux, France) at 5000 rpm. Samples were homogenised for 3 rounds of 30 seconds, placed on ice for 5 min to prevent excess heat generation, and then homogenised again for 3 rounds of 30 seconds before sonication for 5 min. Protein content was quantified using an EZQ Protein Quantitation kit (Thermo Fisher Scientific). For protein visualisation by SDS-PAGE, 10 μg of protein from each sample was mixed with 5 μl of 4X lithium dodecyl sulfate and 50 mM dithiothreitol before heating at 95°C for 5 min. Each sample was then loaded into a pre-cast NuPAGE™ 4 to 12% Bis-Tris polyacrylamide gel (Thermo Fisher Scientific). Gels were run at 180 V for 60 min and fixed briefly before staining overnight with Coomassie Brilliant Blue G-250 (Sigma Aldrich, Burlington, USA). Upon de-staining with Milli-Q water, the gels were imaged on a GelDoc Imaging System (Bio-Rad, Hercules, USA).

All experiments were approved by The University of Adelaide Animal Ethics Committee (approval no. S-2020-063) and by the University of South Australia Animal Ethics Committee (permit no. U24-19), where they were carried out in accordance with the Australian code for the care and use of animals for scientific purposes. The Department for Environment and Water in South Australia gave informed, written consent in the form of a permit (MR00149-1) for conducting sampling in marine protected areas. The commercial fishery also gave informed, verbal consent to conduct collaborative research fishing under their research permit (MRP004).

### Optimisation of wild specimens

**Sample collection and preservation.** *Octopus berrima* were collected directly from a commercial fishery from remote locations in South Australia as previously described [22]. Octopus were euthanised in an ice slurry onboard a small, commercial vessel and transported to land. Subsequently, one arm tip from each octopus was dissected, rinsed with phosphate-buffered saline, and preserved in 5X volume of homemade RNA*later*. Depending on sampling location and travel times to a field location for dissection, tissues were preserved in RNA*later* immediately (sampling from 0 to 30 min and classed as the control group), 3 hours (sampling from

two to three hours), or 6 hours (sampling from five to six hours) after euthanasia. Preserved samples were then stored at 4˚C overnight as recommended by the manufacturer's instructions of commercial RNA*later* (Thermo Fisher Scientific) before being frozen at -20˚C in the field for a duration less than 24 hours. After transportation to the laboratory, tissues were stored at -80˚C until analysis.

**Sample preparation and mass spectrometry.** Tissues were prepared according to the four proposed comparisons (Fig 1). As skin is tougher than the underlying tissues and may contain abundant proteins that make detection of lower-abundant proteins difficult, we investigated if skin influenced the extraction efficiency and dynamic range using only the $LN_2$ homogenisation method by comparing tissues with and without skin. We also compared the efficiency of two common tissue homogenisation methods, $LN_2$ and metal beads, using only samples without skin due to the limited amount of sample available. Lastly, using an optimised approach based on our results from comparing sexes, tissue types and tissue homogenisation methods, we then compared the time delay in RNA*later* preservation using only samples without skin and homogenised using metal beads (Fig 1). Four octopuses were used in the comparisons of sex, tissue types and tissue homogenisation methods, whereas nine different octopuses were used for the comparison of time delay in preservation.

Tissues (approximately 5 mg) were homogenised by either immersion in $LN_2$ and crushing into a powder using a mortar and pestle, or by combining metal beads (7X tissue weight) and 375 µl homemade RIPA buffer (S1 Table) in a tube using the Bullet Blender Storm 24 (Next Advance, Troy, USA) for tough muscular tissues such as octopus arms. The latter consisted of 1 min bead-beating and 30 seconds of cooling to prevent excess heat generation, which was repeated three more times before centrifugation for 30 minutes at 4˚C, 20000 ×*g*. The supernatant was then transferred to cold acetone (6X volume of tissue weight for $LN_2$; 4X for beads; Chem-Supply, Port Adelaide, Australia) and stored overnight at -20˚C. Upon centrifugation (4˚C, 20000 ×*g*), the pellets were air-dried and dissolved in 8 M urea (Merck, Darmstadt,

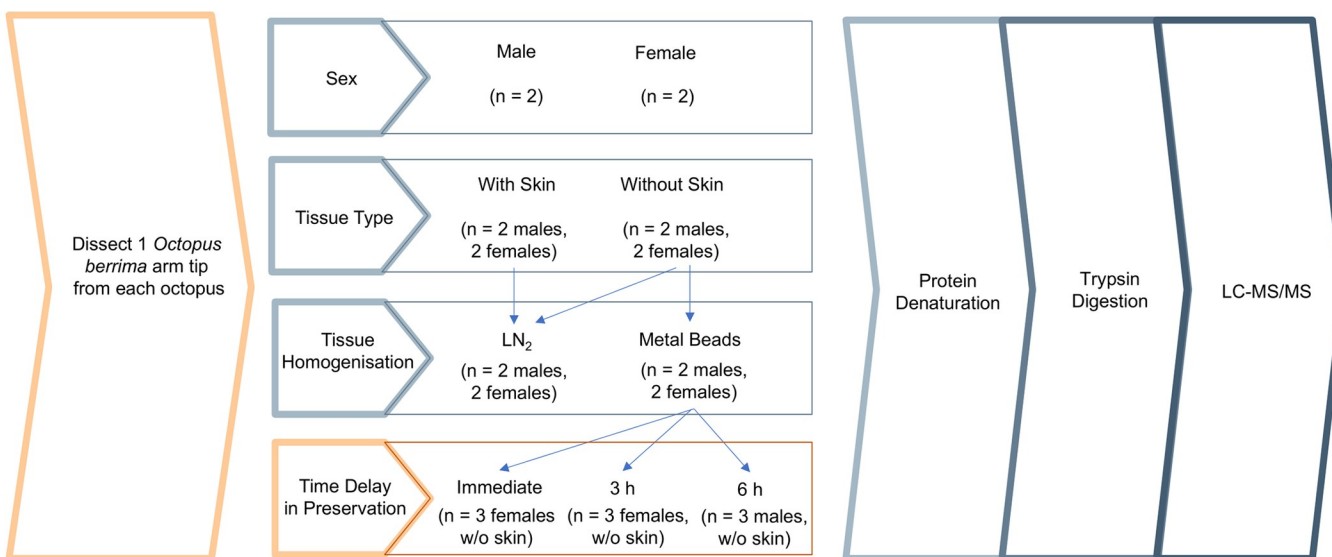

**Fig 1. Workflow for the optimisation of wild specimens comprising field (orange) and lab (blue) work for proteomic analyses.** Four types of comparisons were made using *Octopus berrima* arm tips: sex (male and female), tissue types (with and without skin), homogenisation methods of tissues ($LN_2$ and metal beads) as well as immediate or delayed tissue preservation [immediate (control), 3 h, and 6 h]. Tissue types were compared using the $LN_2$ homogenisation method, whereas tissue homogenisation methods were compared using only samples without skin. Using an optimised approach based on our results, time delay in preservation was then compared using only samples without skin and homogenised using metal beads. All samples irrespective of comparison type underwent protein denaturation and digestion before liquid chromatography-tandem mass spectrometry (LC-MS/MS) was performed.

Germany) with 50 mM ammonium hydrogen carbonate (Merck) and sonicated twice for a minute. Protein was quantified using an EZQ Protein Quantitation kit (Thermo Fisher Scientific). The solution was combined with 10 mM dithiothreitol (Roche, Basel, Switzerland) for 45 minutes at room temperature followed by the addition of 20 mM iodoacetamide (Honeywell, Charlotte, USA) for a 30 minute in-the-dark incubation at room temperature. Urea was diluted to 1 M using 50 mM ammonium hydrogen carbonate, with 1 μg trypsin (Promega, Madison, USA) added to 50 μg protein for overnight incubation at 37˚C. The digestion was halted by formic acid (Sigma Aldrich, Burlington, USA). Upon peptide clean-up using ZipTips (Merck, Co Wicklow, Ireland), 1 μg of peptide sample was analysed by LC-MS/MS on an Ultimate 3000 RSLCnano or EASY-nLC 1200 system connected to an Orbitrap Exploris 480 mass spectrometer (Thermo Fisher Scientific). Peptides were resuspended in 0.1% formic acid and loaded onto a 25 cm fused silica column (75 μm inner diameter, 360 um outer diameter) heated to 50˚C. The column was packed in-house with 1.9 um ReproSil-Pur 120 C18-AQ particles (Dr. Maisch, Ammerbuch, Germany). Peptide separation was conducted over a 70-minute linear gradient (3 to 20% acetonitrile in 0.1% formic acid) at a flow rate of 300 nl/min. Compensation voltages (-50 and -70 V) were applied from a FAIMS Pro interface (Thermo Fisher Scientific) to filter the entry of ionised peptides into the mass spectrometer. For each compensation voltage, the cycle time was limited to 1.5 seconds and the dynamic exclusion period set to 40 seconds. MS scans ($m/z$ 300 to 1500) were measured at resolution 60000 ($m/z$ 200) in positive ion mode. MS/MS scans were acquired at resolution 15000 in a data-dependent manner (minimum threshold of $1 \times 10^4$ precursor ions), with peptide fragmentation performed with 27.5% higher-energy collision dissociation.

**Data analysis.** To ensure statistical independence, bioinformatics analyses were conducted one comparison at a time using separate datasets differing by only one variable (e.g. sex). Raw files were processed with Proteome Discoverer 2.5 software (Thermo Fisher Scientific), using the Sequest HT search engine and a UniProt unreviewed "Cephalopoda" database (version 2021_03; 73232 protein entries). Carbamidomethylation of cysteine was set as a fixed modification, while variable modifications consisted of methionine oxidation, N-terminus acetylation, N-terminus loss of methionine and N-terminus loss of both methionine and acetylation. Other parameters in this workflow were at default settings i.e. maximum two missed cleavages, with a 10 ppm precursor mass tolerance and 0.1 Da fragment mass tolerance. Peptide group abundances were based on the intensity of their precursors, while protein abundances were calculated as the summation of their associated peptide group abundances. Protein ratio calculations were based on pairwise ratios using ANOVA, with other parameters at default settings i.e. protein and peptide false discovery rates were set to 0.01, peptide confidence was set to at least 'High' and at least one peptide sequence was required to identify a protein.

This analysis was repeated for other tissue comparisons, with each protein list filtered for Master proteins possessing a 'High' protein false discovery rate confidence to obtain totals of unique proteins for every comparison. To determine if experimental conditions in each comparison were significantly different, the protein lists were further filtered with an additional criterion (p-value $\leq$ 0.05 for adjusted abundance ratio) before heat maps (using normalized protein abundances) and volcano plots were generated. All mass spectrometry proteomics data have been deposited to the ProteomeXchange Consortium via the PRIDE [23] partner repository with the dataset identifier PXD043096.

# Results

In order to assess whether RNA*later* would be a suitable medium for tissue storage, we first evaluated protein integrity using SDS-PAGE (Fig 2). Raw gel images can be found in S1 Raw

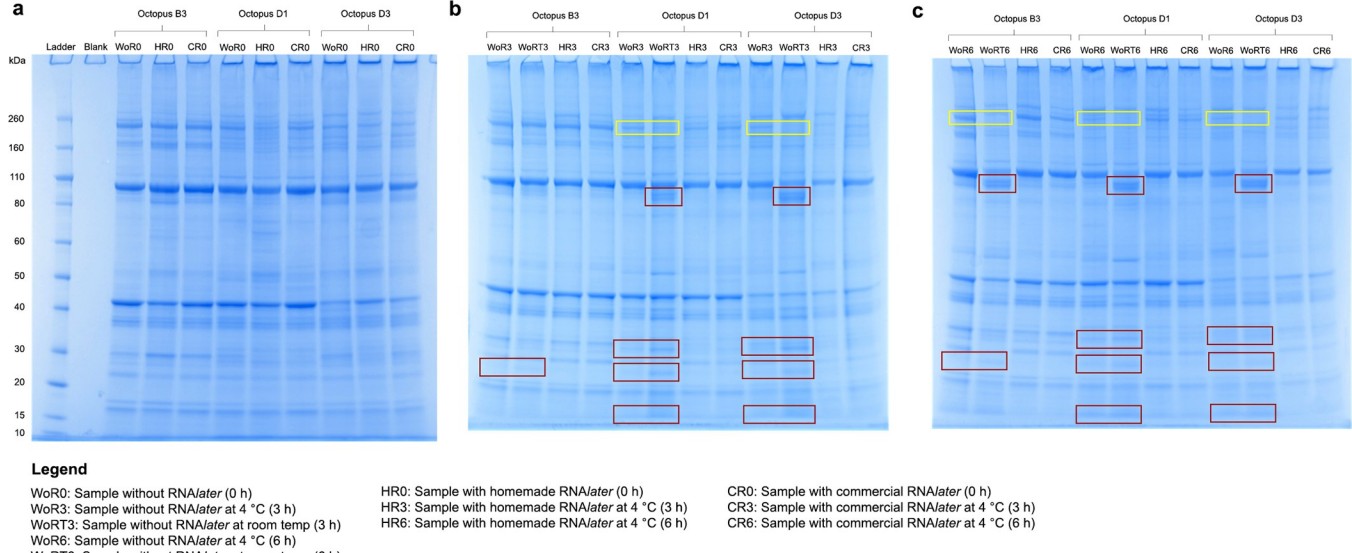

**Fig 2. SDS-PAGE gels comparing the influence of RNA*later* on *Octopus berrima* proteins.** *Octopus berrima* tissues without RNA*later* were compared with those incubated in homemade and commercial RNA*later* for three durations: 0 h (a), 3 h (b) and 6 h (c). For samples without RNA*later*, tissues stored at 4°C were compared with those stored at room temperature. Red boxes indicate higher band intensities for lower-mass proteins while yellow boxes indicate lower band intensities for higher-mass proteins, suggesting higher levels of protein degradation in samples without RNA*later*.

images. The higher band intensities for lower-mass proteins and the lower abundance of higher-mass proteins in samples without RNA*later* compared to those with RNA*later* suggest higher levels of protein degradation occurred when tissues were not preserved in RNA*later*. This pattern was more pronounced at the 6 h timepoint, even when samples without RNA*later* were stored at 4°C. There were also higher band intensities for lower-mass proteins in samples without RNA*later* stored at room temperature compared to those stored at 4°C, which were prominent at the 3 h and 6 h timepoints, suggesting higher amounts of degraded proteins were present after room temperature storage. Lastly, as RNA*later* can be made in-house or bought commercially at a high cost, we also wanted to determine if homemade RNA*later* was suitable for proteomic analyses. We showed that homemade RNA*later* worked as well as the commercial RNA*later* due to similar band intensities and thereby the same negligible protein degradation. Based on these findings, we used cold storage and/or homemade RNA*later* for the analysis of wild-caught specimens in subsequent proteomic experiments.

In order to optimise in-lab methods for tough muscular tissues such as in octopus, we examined the effect of different conditions (tissue types, sex, and tissue homogenisation methods) on the proteomic analysis of wild-caught octopus. We successfully identified over 3500 protein groups from all tissues (Table 1), where unique protein groups were distinguished by at least one unique peptide. Higher protein numbers were obtained using the metal beads compared to $LN_2$, where there was an increase in the abundance of 672 proteins prepared

**Table 1. Total number of unique proteins identified in the optimisation of wild specimens for comparisons between: Male (M) and female (F), tissues with (WS) and without skin (WoS), tissues homogenisation method ($LN_2$ and metal beads), and immediate and delayed tissue preservation.**

| Experiment | Sex | | Skin | | Tissue Homogenisation | | Time Delay in Preservation (hours) | | |
|---|---|---|---|---|---|---|---|---|---|
| | **M** | **F** | **WS** | **WoS** | **$LN_2$** | **MetalBeads** | **Immed-iate** | **3** | **6** |
| Total number of proteins | 3700 (WoS) | 3527 (WoS) | 3952 | 3750 | 3996 | 4356 | 3708 | 3709 | 3703 |
| | 3868 (WS) | 3829 (WS) | | | | | | | |

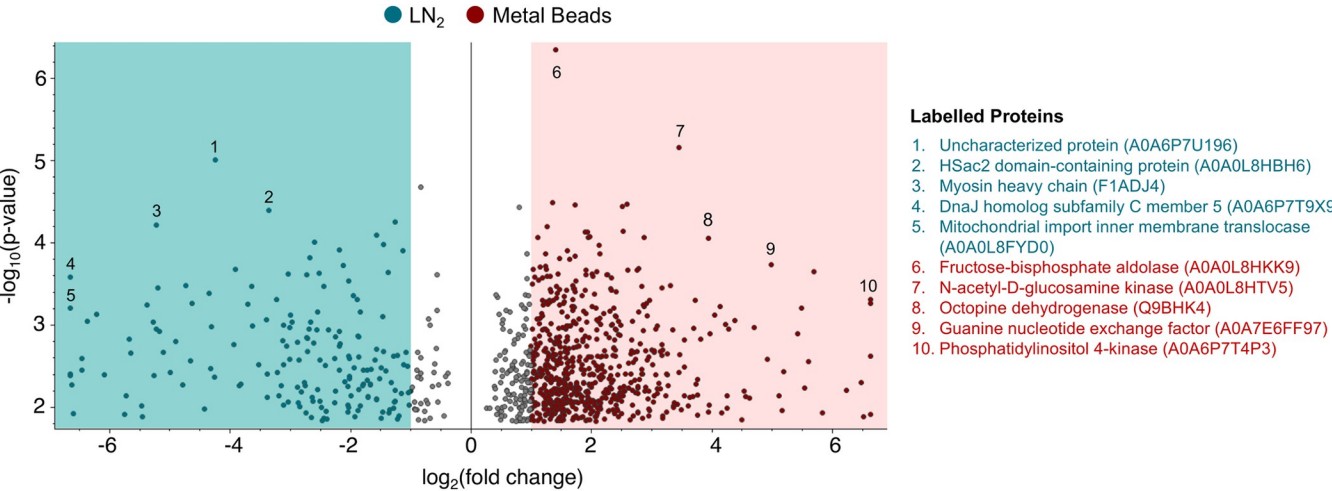

**Fig 3. Significant protein abundance differences between tissue homogenisation methods.** Magnified volcano plot showing greater number of proteins with higher abundance obtained using metal beads (672 proteins) compared to $LN_2$ (175 proteins). Each dot represents one protein. Red dots represent proteins of significantly higher abundance ($p \leq 0.05$; $log_2$ fold change >1) for metal beads while blue dots represent proteins of significantly higher abundance for $LN_2$ ($p \leq 0.05$; $log_2$ fold change <-1). Grey dots represent proteins with $log_2$ fold changes between -1 and 1. Ten proteins with higher significance values and fold changes are labelled.

from metal beads compared to an increase in abundance of 175 proteins prepared from $LN_2$ (Fig 3). Some of the higher abundance proteins from the metal bead method were muscle-specific proteins such as actin, myophilin and tropomyosin (S2 Table). A comprehensive list of proteins which differed significantly in abundance from using metal beads and $LN_2$ is provided in S3 and S4 Tables respectively, whereas a complete list of proteins for this comparison is provided in S5 Table. No significant differences in protein abundance ratios were found between sexes and tissue types. Based on these findings, we conducted the subsequent experiment using metal beads and tissues without skin for the analysis of time delay before tissue preservation.

Lastly, we examined the effect of time delay before tissue preservation on the protein integrity of wild-caught specimens in field settings, which resulted in the identification of over 3700 protein groups from all tissues (Table 1). Although pairwise analyses with ANOVA indicated no significant differences in overall protein abundance ratios found between immediate and delayed preservation, the 6 h time-point replicates in the heat map clustered together (Fig 4). This suggests that while tissue storage in the cold allows for comprehensive proteomic analyses, earlier storage with RNA*later* can more suitably preserve proteome integrity. A complete list of proteins identified in the comparison of time delay in preservation is provided in S6 Table.

## Discussion

Proteomics is a powerful technique in ecological applications, but the lack of optimised protocols for field-collected samples has contributed to its under-utilisation. Our study has provided an optimised approach at various stages of collecting and analysing wild specimens. Our initial findings showed that RNA*later* was effective in slowing protein degradation, even up to the 6 h timepoint where intact proteins were still present and did not hinder protein analyses. This finding is likely due to the high concentrations of quaternary ammonium sulfates that denature proteases (including DNAses and RNAses), thereby preserving proteins and nucleic acids

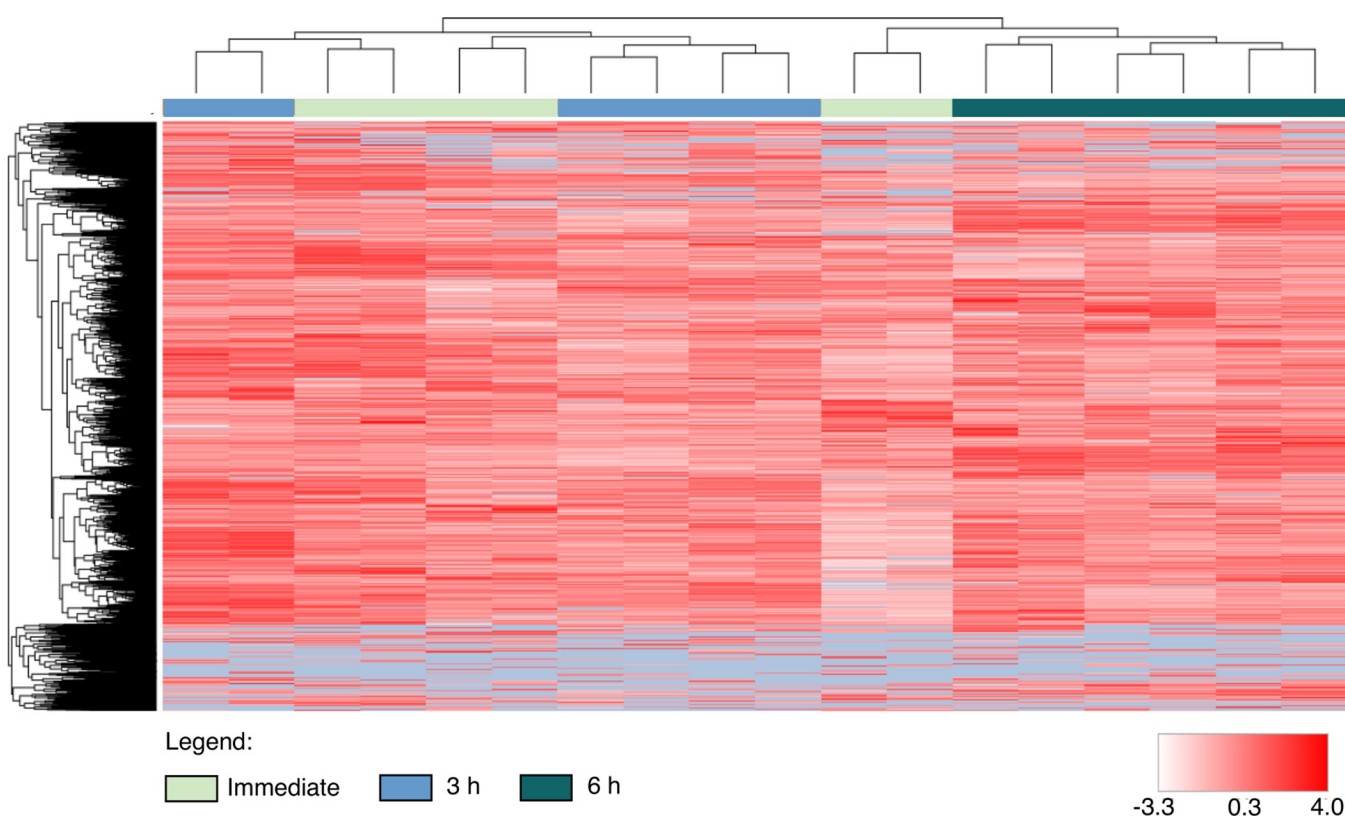

**Fig 4. Protein abundance did not differ between immediate and delayed tissue preservation.** Heat map of identified proteins from immediate (control), 3 h and 6 h delayed tissue preservation (n = 9). The intensity of red reflects protein z-scores (-3.3 to 4.0), with grey denoting absent proteins.

[9, 10]. This result suggests that storing samples in RNA*later* can overcome tissue preservation challenges associated with the lack of infrastructure during remote fieldwork. We also found that homemade RNA*later* was as effective in preserving proteins as commercial RNA*later*. Since the cost of commercial RNA*later* was 5X that of homemade RNA*later*, this finding allows for significant cost savings as has been the case for subsequent analyses in this study. This makes RNA*later* an ideal storage reagent for other ecological proteomics studies that require the examination of wild specimens.

Over 3500 proteins were successfully identified from octopus arm tips collected from remote field locations using an optimised sample preservation and preparation workflow. Since similar protein abundances and numbers were obtained between immediate and delayed tissue preservation, this indicates that the inevitable delays in tissue preservation associated with challenging field work conditions do not hinder the identification and quantification of thousands of proteins. Taking results from both experiments together, proteins can be successfully preserved and identified with the use of ice and RNA*later*, indicating that these should be the minimum materials required to conduct proteomic analyses. We note that samples from the in-field experiment could only be preserved on ice in the boat before being preserved in RNA*later* on land, and it is likely that protein degradation started to occur especially after 6 h on the boat, as observed by the slightly higher band intensities from the in-lab experiment for samples stored at 4°C without RNA*later*. Although this suggests that protein degradation is inevitable under in-field collections where only ice is available and that care should be taken to limit the amount of delay for tissue preservation, the low level of protein degradation and the

eventual use of RNA*later* still permitted an extensive number of proteins to be identified and quantified. This is further supported by the fact that no significant differences were found in the protein abundances between immediate and delayed preservation. Moreover, samples in this study were obtained from real-life, remote field conditions in confined, moving spaces where only ice and/or RNA*later* were practical to use. Hence, there is further scope for applying proteomics to a broader range of ecological situations, particularly in cases where tissue preservation may be delayed up to 6 h. However, the long-term protective capabilities of RNA-*later* on proteins remains to be established.

The use of metal beads to homogenise octopus arms yielded a higher number of proteins than $LN_2$, and the significant increase in abundance of muscular proteins indicate that these metal beads are more efficient in breaking up muscular tissues. Bead-beating has been a traditional method of tissue homogenisation used to disrupt the tissue matrix in order to release nucleic acids and/or proteins [24]. This finding may be applicable to other molluscan tissues which may be tougher to homogenise than softer tissues, such as fish and soft-bodied insects.

Although tissues with and without skin did not exhibit significant differences, various factors must be considered when deciding to include skin in analyses. In cases where skin proteins form the research topic or where time is lacking, the extra effort in dissecting and removing skin may not be worthwhile, but one must ensure consistent proportions of skin to muscle to enable fair comparisons. Otherwise, as consistent skin to muscle ratio can also be challenging in terms of reproducibility, skin removal should be done for all tissues.

Finally, the lack of significant protein differences between the sexes could be due to the same section of arm muscle tissue being collected and the arm tip samples being anatomically similar. In addition, male-specific mating arms (hectocotylus) were not used in any of the comparisons. Future reproductive studies in cephalopods could investigate proteomic differences between sex-specific arms.

A major limitation of proteomics is the reliance on reference genome information, as confident protein identifications are highly dependent on sequence databases [25], which may not be readily available for uncommon organisms. The ability to obtain more physiologically relevant information is also dependent on how extensively the databases are annotated, as well as reliability of the curation itself. While several octopus genomes have been sequenced [26–29], the species in this study, *Octopus berrima*, has not been sequenced. Even within the current database for cephalopod proteins, many functional annotations are missing, and numerous proteins remain unreviewed, hence this could limit the usefulness of proteomic analyses. To minimise this limitation, a cross-species, homologous protein sequence database within Cephalopoda from UniProt was used to identify proteins in our samples. However, reliable proteomic analyses will still depend on factors beyond our control as future studies continue working on creating curated protein databases. Nevertheless, other cephalopods could potentially be studied by proteomic analyses in the future using databases such as UniProt.

In summary, our optimised protocol assists the development of proteomics research in unique ecological applications. We have demonstrated the usefulness of ice and RNA*later* in preserving proteins, making it safe and suitable for many ecological applications. We have also shown that extensive proteomic coverage can be obtained even when challenging field sampling conditions result in delayed tissue preservation, provided the samples are stored in ice and RNA*later*. We also recommend the use of metal beads, especially when homogenising tough tissues because of the higher protein numbers obtained with this method. It is worth noting that with ecological proteomics still in its infancy, reliably curated and annotated genomes for various organisms of interest remain necessary to facilitate protein identifications.

## Supporting information

**S1 Table. Recipe for homemade RNA*later* and RIPA buffer.**
(PDF)

**S2 Table. List of muscular proteins that were significantly higher in abundance in samples homogenised using metal beads compared to homogenisation with LN$_2$.**
(PDF)

**S3 Table. List of proteins that were statistically significantly higher in abundance in samples homogenised using metal beads compared to homogenisation with LN$_2$.** Adjusted p-values can be found in column AB.
(XLSX)

**S4 Table. List of proteins that were statistically significantly higher in abundance in samples homogenised with LN$_2$ compared to homogenisation using metal beads.** Adjusted p-values can be found in column AB.
(XLSX)

**S5 Table. Complete list of proteins identified from samples homogenised with metal beads or LN$_2$.** Adjusted p-values can be found in column AB.
(XLSX)

**S6 Table. List of proteins identified from immediate (control), 3 h and 6 h delayed tissue preservation.** Adjusted p-values can be found in columns AC, AD and AE.
(XLSX)

**S1 Raw images.**
(PDF)

## Acknowledgments

We are extremely grateful to Leon van Weenen and Lily Perone (SA Premium Octopus) for assistance in octopus collection, as well as Chris Cursaro (Adelaide Proteomics Centre), Enzo Huang (Thermo Fisher Scientific), and Brooke Dilmetz (Mass Spectrometry and Proteomics Facility) for their technical assistance and expertise, without which this project would not be possible.

## Author Contributions

**Conceptualization:** Qiaz Q. H. Hua, Bronwyn M. Gillanders, Zoe A. Doubleday.

**Data curation:** Qiaz Q. H. Hua.

**Formal analysis:** Qiaz Q. H. Hua, Clifford Young.

**Funding acquisition:** Peter Hoffmann, Zoe A. Doubleday.

**Investigation:** Qiaz Q. H. Hua.

**Methodology:** Qiaz Q. H. Hua.

**Project administration:** Qiaz Q. H. Hua, Clifford Young, Tara L. Pukala, Bronwyn M. Gillanders, Zoe A. Doubleday.

**Resources:** Qiaz Q. H. Hua, Clifford Young, Tara L. Pukala, Jasmin C. Martino, Peter Hoffmann, Bronwyn M. Gillanders, Zoe A. Doubleday.

**Supervision:** Bronwyn M. Gillanders, Zoe A. Doubleday.

**Validation:** Qiaz Q. H. Hua, Clifford Young, Tara L. Pukala, Peter Hoffmann, Bronwyn M. Gillanders, Zoe A. Doubleday.

**Visualization:** Qiaz Q. H. Hua.

**Writing – original draft:** Qiaz Q. H. Hua.

**Writing – review & editing:** Qiaz Q. H. Hua, Clifford Young, Tara L. Pukala, Jasmin C. Martino, Bronwyn M. Gillanders, Zoe A. Doubleday.

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
