## [Decision Letter · Decision Letter 0]

20 Apr 2023

PONE-D-23-05036Better late than never: optimising the proteomic analysis of field-collected organismsPLOS ONE

Dear Dr. Hua,

Thank you for submitting your manuscript to PLOS ONE. After careful consideration, we feel that it has merit but does not fully meet PLOS ONE’s publication criteria as it currently stands. Therefore, we invite you to submit a revised version of the manuscript that addresses the points raised during the review process.

We look forward to receiving your revised manuscript.

Kind regards,

Anita Mitico Tanaka-Azevedo

Academic Editor

PLOS ONE

Journal Requirements:

Reviewers' comments:

Reviewer's Responses to Questions

**Comments to the Author**

1. Is the manuscript technically sound, and do the data support the conclusions?

Reviewer #1: Partly

Reviewer #2: Yes

2. Has the statistical analysis been performed appropriately and rigorously? 

Reviewer #1: I Don't Know

Reviewer #2: I Don't Know

3. Have the authors made all data underlying the findings in their manuscript fully available?

Reviewer #1: No

Reviewer #2: Yes

4. Is the manuscript presented in an intelligible fashion and written in standard English?

Reviewer #1: Yes

Reviewer #2: Yes

5. Review Comments to the Author

Reviewer #1: Authors reported a study on the optimization of sample collection and processing for proteomic analysis of field-collected octopus. The study design is straight-forward and is relevant to answer the research question. However, there are a number of issues which compromise the quality of the paper and authors should attempt to address these to satisfaction before the manuscript can be considered for publication.

1. Title: it should be field-collected octopus -- as appeared in the title stated in other parts of the submission (supplementary files). Authors only performed the work on octopus, not all representative organisms.

2. I understand that authors wanted to study the "proteomics" of "octopus" using the octopus tissues which were stored and processed differently (hence the optimizing methods). However authors need to be clear in the beginning and state in the introduction what "proteomes" are they looking for, and subsequently the justification of the materials and techniques applied.

3. Other suggestions for introduction:

Line 77: what is traceability of seafood - this seems a bit out of place in the context of this statement.

Also, in the context of animal studies, proteomics has been widely applied in snakes to unravel the venom complexity and diversity including intra- and inter-species variation, which carries ecological, evolutionary and medical significance. This part of information should be included. Suggest references (some examples of reviews on snake venomics):

- https://www.frontiersin.org/articles/10.3389/fphar.2021.768015/full

- https://www.mdpi.com/2072-6651/14/4/247

- https://www.ncbi.nlm.nih.gov/pmc/articles/PMC5408369/

Line 80-81: ... proteomics-based in ecology - please be specific regarding which aspect of ecology studies authors feel that proteomics is lacking?

Line 97-99: storage solutions - what about undenatured ethanol? This is a very commonly and easy-to-use method for tissue sampling in the wild. Authors should include this in the introduction and provide justification.

4. Methods:

Line 136: How long were the octopus were kept and raised in aquaria?

Line 140-141: .......arm sections were dissected and immediately stored at -20 C... was this in the presence of RNAlatere? What is the ratio of RNAlater to the tissue?

Line 142... maintained at 4 C. with those at room temperature without RNAlater... the description is unclear. What is the volume, duration, amount of tissue etc.... these technical information needs to be included and described clearly.

Section: Proteomics

- How did authors define "protein abundances"? Do authors mean relative fold-change in "protein expression"? How was this measured, is it by the number of proteins identified per sample, or by the amount of protein (in terms of relative weight) per sample? Please provide the quantitative determination of the protein abundance for each proteome.

As a common practice in sequencing and proteomics/genomics study, the raw data of MS should be deposited and archived in a repository for public access. If this is available, please provide the repository accordingly.

Results and Discussion:

Provide the statistical significance of all parameters used in comparison, in tables and figures.

Suggest to include limitations of the work.

Reviewer #2: Dear authors, thank you for the opportunity to review such a clearly and convincingly written paper. Unfortunately, I am an ecologist and am far from qualified to judge whether the chemistry or bioinformatics were appropriate for the task. Indeed, probably half of the paper, from the methods through the results could have been completely made up and I wouldnt be able to tell. However, given the clarity of the writing and knowing that Zoe Doubleday and Bronwyn Gillanders were part of the research and the paper, I fully trust that this was the not case.

I had a couple of questions/comments:

I might have missed it (or misunderstood), in which case it could be made clearer, but I was a little uncertain as to what the RNAlater preserved samples were compared against. This confusion arose somewhere around the first sentence of the methods. I understood that they were compared against samples that were not treated with RNAlater, but wouldnt it have made most sense to also understand how they performed against the current lab best-practice for handling of samples for proteomic analyses, presumably freezing immediately at -80C (or being treated with some other less-transportable preservative)? Its not a deal-breaker but might be worth clarifying for a non-proteomics expert if/where you think it would be appropriate in the paper.

Very minor detail: the wording of the sentence L86-88 is a little strange and sounds like the sampling sites are inside a moving car. It would be worth adjusting the wording there slightly.

Other than that, well done!, and I hope that the second reviewer had some knowledge in proteomics... or at least chemistry... or bioinformatics... to be able to give you feedback on those sections of the manuscript.

All the best...

6. PLOS authors have the option to publish the peer review history of their article (what does this mean?). If published, this will include your full peer review and any attached files.

Reviewer #1: No

Reviewer #2: No

---

## [Author Response · Author response to Decision Letter 0]

13 Jun 2023

We would like to thank the editor and reviewers for their valuable feedback. Please find our specific responses in the uploaded documents ("Response to Reviewers" and "New Cover Letter").

---

## [Editor Report · Decision Letter 1]

19 Jun 2023

Better late than never: optimising the proteomic analysis of field-collected octopus

PONE-D-23-05036R1

Dear Dr. Hua,

We’re pleased to inform you that your manuscript has been judged scientifically suitable for publication and will be formally accepted for publication once it meets all outstanding technical requirements.

Kind regards,

Anita Mitico Tanaka-Azevedo

Academic Editor

PLOS ONE

---

## [Editor Report · Acceptance letter]

4 Jul 2023

PONE-D-23-05036R1 

­­Better late than never: optimising the proteomic analysis of field-collected octopus 

Dear Dr. Hua:

I'm pleased to inform you that your manuscript has been deemed suitable for publication in PLOS ONE. Congratulations! Your manuscript is now with our production department. 

Kind regards, 

on behalf of

Dr. Anita Mitico Tanaka-Azevedo 

Academic Editor

PLOS ONE